# Chances of Mortality Are 3.5-Times Greater in Elderly Patients with Umbilical Hernia Than in Adult Patients: An Analysis of 21,242 Patients

**DOI:** 10.3390/ijerph191610402

**Published:** 2022-08-21

**Authors:** Saral Patel, Abbas Smiley, Cailan Feingold, Bardia Khandehroo, Agon Kajmolli, Rifat Latifi

**Affiliations:** 1Westchester Medical Center, School of Medicine, New York Medical College, Valhalla, NY 10595, USA; 2Minister of Health, Republic of Kosova, Adjunct Professor of Surgery, University of Arizona, Tucson, AZ 10000, USA

**Keywords:** umbilical hernia, mortality, hospital length of stay, elderly

## Abstract

The goal of this study was to identify risk factors that are associated with mortality in adult and elderly patients who were hospitalized for umbilical hernia. A total of 14,752 adult patients (ages 18–64 years) and 6490 elderly patients (ages 65+), who were admitted emergently for umbilical hernia, were included in this retrospective cohort study. The data were gathered from the National Inpatient Sample (NIS) 2005–2014 database. Predictors of mortality were identified via a multivariable logistic regression, in patients who underwent surgery and those who did not for adult and elderly age groups. The mean (SD) ages for adult males and females were 48.95 (9.61) and 46.59 (11.35) years, respectively. The mean (SD) ages for elderly males and females were 73.62 (6.83) and 77.31 (7.98) years, respectively. The overall mortality was low (113 or 0.8%) in the adult group and in the elderly group (179 or 2.8%). In adult patients who underwent operation, age (OR = 1.066, 95% CI: 1.040–1.093, *p* < 0.001) and gangrene (OR = 5.635, 95% CI: 2.288–13.874, *p* < 0.001) were the main risk factors associated with mortality. Within the same population, female sex was found to be a protective factor (OR = 0.547, 95% CI: 0.351–0.854, *p* = 0.008). Of the total adult sample, 43% used private insurance, while only 18% of patients in the deceased population used private insurance. Conversely, within the entire adult population, only about 48% of patients used Medicare, Medicaid, or self-pay, while these patients made up 75% of the deceased group. In the elderly surgical group, the main risk factors significantly associated with mortality were frailty (OR = 1.284, 95% CI: 1.105–1.491, *p* = 0.001), gangrene (OR = 13.914, 95% CI: 5.074–38.154, *p* < 0.001), and age (OR = 1.034, 95% CI: 1.011–1.057, *p* = 0.003). In the adult non-operation group, hospital length of stay (HLOS) was a significant risk factor associated with mortality (OR = 1.077, 95% CI: 1.004–1.155, *p* = 0.038). In the elderly non-operation group, obstruction was the main risk factor (OR = 4.534, 95% CI: 1.387–14.819, *p* = 0.012). Elderly patients experienced a 3.5-fold higher mortality than adult patients who were emergently admitted with umbilical hernia. Increasing age was a significant risk factor of mortality within all patient populations. In the adult surgical group, gangrene, Medicare, Medicaid, and self-pay were significant risk factors of mortality and female sex was a significant protective factor. In the adult non-surgical group, HLOS was the main risk factor of mortality. In the elderly population, frailty and gangrene were the main risk factors of mortality within the surgical group, and obstruction was the main risk factor for the non-surgical group.

## 1. Introduction

First cited in 1808, umbilical hernias (UH) are trackable by Pubmed in a total number of 4716 scientific articles over the past two centuries [1]. About 6% to 14% of all abdominal wall hernias among adults are UH [2] with an overall prevalence of 2% within the adult population [3]. Additionally, 90% of adult UH are acquired as the result of increased pressure within the abdomen and have predisposing factors such as obesity, a history of multiple pregnancies with prolonged labor, ascites, and intra-abdominal tumors [4]. It is believed that adipose tissue likely separates layers of muscle, which causes the abdominal musculature to stretch, allowing for the formation of umbilical hernias [5]. UH typically present asymptomatically in the setting of a small hernia though pain, tenderness, and GI discomfort; strangulation or incarceration can be seen in patients with larger hernias [5]. The gender stratification of UH seems to be disputed with some studies indicating UH are more common in women [4] while others indicate they are more common in men [6]. It is generally agreed that the geriatric population faces a larger risk of developing UH [7] which is attributed to the weakening of abdominal muscles [8]. While there is insight into risk factors of UH development, the literature on predictors of mortality is scarce. Considering the prevalence of UH, it is essential that this condition and its associated risk factors for mortality be investigated. This study aimed to shed light on a wide range of contributors that suggest increased odds of mortality for adult and geriatric individuals emergently admitted for UH in the National Inpatient Sample (NIS) 2005–2014.

## 2. Materials and Methods

This study was a retrospective cohort study which utilized the National Inpatient Sample (NIS) 2005–2014 data set to extract data on adult (ages 18–64 years) and elderly (ages 65+ years) patients emergently admitted for UH. The NIS data set was generated as a part of the Healthcare Cost and Utilization Project (HCUP), which aims to create uniform data on patients within a particular population. The Agency for Healthcare Research and Quality (AHRQ) sponsors HCUP. The International Classification of Diseases (ICD-9) codes, based on the World Health Organization’s Ninth Revision, used to identify patients with umbilical hernia were 551.1 (UH with gangrene), 552.1 (UH with obstruction), and 553.1 (UH without obstruction or gangrene) [9,10,11]. The characteristics that were obtained and analyzed of patients and hospitals are as follows: age, gender, race, income quartile, primary diagnosis, hospital location, associated comorbidities (deficiency anemias, congestive heart failure, chronic pulmonary disease, uncomplicated diabetes, hypertension, peripheral vascular disorder, fluid and electrolyte disorders, solid tumor, metastatic cancers, and weight loss), presence of intestinal gangrene or obstruction, health insurance status (Medicare, Medicaid, private insurance, self-pay, and no charge), invasive diagnostic status, surgical procedure status, days to first procedure, hospital length of stay (HLOS), total charges, and modified frailty index. Those in income quartile 1 had the lowest incomes while those in income quartile 4 had the highest incomes. The 5-item modified frailty index was calculated based on the patient status of the following five items: diabetes mellitus, hypertension, chronic pulmonary disease, congestive heart failure, and functional health status. Because functional health status was not included in the NIS data, it was estimated with the assumption that the presence of renal failure, cancer, paralysis, coagulopathy or weight loss was considered as partial or total loss of functional health status. The index ranged from 0 to 5, with 0 as the least frail and 5 as the frailest. The ICD-9 codes of invasive procedures and operations can be found in Table 1.

### Statistical Analysis

Descriptive statistics were used to present the numerical findings. For each numerical variable, the mean, standard deviation (SD), and confidence interval at 95% (CI) were calculated. Categorical variables were compared with chi-square tests and continuous variables were analyzed with *t*-tests. Binary multivariable logistic regression analysis and multivariable generalized additive model (GAM) were used to evaluate the role of different factors in anticipating mortality. Adjustments were made for both these multivariable models for the following attributes of patients and hospitals: age, sex, race, income quartile, health care insurance, hospital location, modified 5-item frailty index, invasive diagnostic procedures, hernia complications and time to operation. The predicted smooth functions along with the confidence intervals were plotted in multivariable GAM models. A *p* value was considered significant if it was less than 0.05. SPSS software version 24 (SPSS Inc., Chicago, IL, USA) and R statistical software (Foundation for Statistical Computing, Vienna, Austria) were used to formulate all analyses.

## 3. Results

### 3.1. Sex Categories

There were 14,752 adults (ages 18–64 years) who were emergently admitted with the primary diagnosis of UH that were included in the current study. Within this population, there were 8616 males and 6136 females. There were also a total number 6490 elderly patients (ages 65+) with 2843 males and 3647 females. A total of 56.2% of elderly patients were female, while only 41.6% of adults were female. The first stratified analysis based on sex categories is presented in Table 2.

For the adults, there were sex differences for race, income quartiles, insurance, comorbidities, and complications. The same differences were noted in elderly patients, apart from income quartile. Quartile 1 for income was the most common for all patients and Quartile 4 was the least common one. For both elderly and adult male populations, the following comorbidities were more likely to be seen: alcohol abuse, coagulopathy, drug abuse, liver disease, peripheral vascular disorders, renal failure, and solid tumors. The comorbidities more commonly observed in female elderly and adult populations, compared to the respective male groups, included: rheumatoid arthritis, depression, uncomplicated diabetes, hypothyroidism, and obesity. Additionally, adult males were more likely to have AIDS, congestive heart failure, lymphoma, and weight loss, compared to adult females. Adult females were more likely to have chronic blood loss, chronic pulmonary disease, metastatic cancer, and psychoses than their male counterparts. Within the elderly population, males had more cases of chronic pulmonary disease, and females were more likely to have hypertension, fluid/electrolyte disorders, and neurological disorders (Table 2). Adult males had a longer time to surgery while elderly males had a shorter time to surgery, when compared to their female counterparts. Higher total charges were noted in both adult and elderly males.

### 3.2. Mortality

Stratified analysis based on outcome categories (survived vs. deceased) is shown in Table 3. For the adult population, 113 (0.8%) of the patients expired. The mean (SD) age for the adult deceased population was 54.05 (7.25) years while that of the surviving population was 47.9 (10.44) years. For the elderly deceased population, the mean age was 78.6 (8.5) years while the surviving population’s mean age was 75.61 (7.68) years (Table 3). Elderly patients experienced an overall 3.5-times greater mortality rate than adult patients.

Men experienced higher mortality in the adult group, while there was no significant sex difference in mortality for the elderly population (Table 2). A total of 41.6% of the adults in this study were female, while they made up only 25.7% of the adult deceased population. In the adult population, Medicare, Medicaid and self-pay were seen more in the deceased group when compared with private insurance. Significant differences for insurance status were not noted in the elderly population (Table 3). Within both adult and elderly populations, deceased patients were more likely to have coagulopathy, liver disease, fluid/electrolyte disorders, metastatic cancer, neurological disorders, renal failure, solid tumor, and weight loss. The only comorbidity seen more in surviving members of either age group was obesity. In the adult population, comorbidities seen more among the deceased were AIDS, alcohol abuse, deficiency anemias, peripheral vascular disorders, psychoses, and pulmonary circulation disorders. In the deceased elderly population, the more common comorbidities were CHF, paralysis, and valvular disease. In the survived elderly population, the more common comorbidities were depression and hypertension (Table 3).

About 15–20% of the total adult population did not experience UH complications compared to only 10.6% of the deceased adult patients. Similarly, about 13% of the total elderly population did not have UH complications, while only 4.5% of the elderly deceased group saw a similar course. Obstruction was the most common UH complication for both survived and deceased groups in both ages. The Modified Frailty Index was higher in the deceased population for both age groups, when compared to that of the total population. Time to first surgical procedure was higher in deceased patients in the adult group, while no significant differences were noted in the elderly population. HLOS was notably higher for deceased patients of both age groups. Lastly, total charges were higher in deceased patients for both age groups (Table 3).

### 3.3. Operation vs. No Operation

Surgical or invasive procedures were seen more in males in both age groups (Table 2). The stratified analysis based on surgery status is presented in Table 4. In the adult population, 86.3% of patients underwent surgery, while 13.7% did not have surgery. For the elderly population, 78.9% of patients had surgery, while 21.1% did not. Elderly patients had a higher mortality rate in both the non-surgical and surgical groups (3.1% and 2.7%, respectively), when compared to the adult non-surgical and surgical samples (0.5% and 0.8%, respectively). Income quartile differences based on surgical status were noted in the adult population only. It was more common to see comorbidities in the non-operative group for the adult population, but this relationship was not as significant for surgery status in the elderly population. Patients who were operated on had a higher proportion of gangrene and obstruction than those not operated on. The Modified Frailty Index was higher in the adult non-operative group than that of the operative group and was not found to be significantly different in the elderly surgical and non-surgical population. HLOS and total charges were higher for the surgical group for both adult and elderly populations.

### 3.4. Risk Factors for Mortality

The multivariable logistic regression model for mortality was built separately for both adult and elderly patients who underwent operation (Table 5), and for those who did not (Table 6). Mortality was the dependent variable. Elderly patients were found to have a 2.5-times greater odds ratio for mortality with gangrene and a 1.6-times greater odds ratio for mortality with obstruction than that of the adult population. In patients who underwent operation, age and gangrene were risk factors of mortality for both age groups. Medicare, Medicaid, self-payments, and male sex were the main risk factors for mortality for the adult surgical population. In the elderly surgical population, frailty was a risk factor for mortality. The Modified Frailty Index was 56% higher in the elderly population compared to adults (Table 2). Being an adult female reduced the odds of mortality after surgery by 45.3% (Table 5). Compared to adult operative patients with private insurance, adult operative patients with Medicare were 5.4-times more likely to experience mortality, with a similar increase of 5-times more likely for Medicaid and 4.6-times more likely in self-pay patients. In patients who did not undergo operation, age was the main risk factor for mortality for both age groups. Increasing HLOS was a risk factor for the adult non-surgical group, raising the risk of mortality by 7.7% for each additional day. All elderly patients had a longer average HLOS (5.36 days) than that of the total adult patients (3.56 days) (Table 2). Obstruction was a risk factor for mortality for the elderly non-operation group (Table 6).

## 4. Discussion

The primary aim of this study was to evaluate associations between demographics, socioeconomic status, comorbidities, surgery status, and overall mortality in non-elderly adult and elderly patients undergoing emergency admission for UH. Our findings have demonstrated that age, male sex, increasing HLOS, insurance status, hernia complications, and frailty were the main predictors of mortality in subjects emergently admitted with umbilical hernia. The mortality rate in elderly patients was 3.5-times that of the adult sample, and may be attributed to the weakening of the abdominal wall in elderly patients [8]. HLOS of elderly patients was nearly 2 days longer than that of adults, and the Modified Frailty Index was nearly double in the elderly group. Insurance status was an indicator of mortality in the adult non-operative group, but this did not apply to the elderly population, likely due to the homogenous distribution of payment status for elderly patients.

Recent trends in the United States show a significant rise in the rates of emergent incisional hernia repair in males with increasing age, from 7.8 per 100,000 people in 2001 to 32.0 per 100,000 in 2010, whereas the rates for females have remained stable [12]. The results from our study indicate that adult male patients treated with surgery have higher odds of mortality than females. Specifically, adult males accounted for 58.4% of the adult sample and 74.3% of adult mortalities. Furthermore, in a similar analysis of the National Inpatient Database, 2005–2014, Smiley et al. found male sex to be a risk factor for mortality in adult patients emergently admitted for ventral hernia [13]. The recent evidence for male sex as a risk factor for hernia repair contradicts the older paradigm that claimed the opposite. In 2007, Nilsson et al. found that women face greater mortality after groin hernia repair than men due to the increased risk of emergency procedure [14] although the dissimilarity in this finding may be due to the difference in the type of hernia. Interestingly, 23.6% of the adult men and 14.6% of the elderly men included in this study had liver disease as a comorbidity, compared to only 9.6% of adult females and 7% of the elderly females (Table 2), and liver disease has previously been shown to be a major predictor of mortality in UH patients [15,16].

### 4.1. Increasing Age and Mortality

Increased age was a risk factor of mortality for all patients in this study, which could be attributed to the increased incidence of cardiovascular and pulmonary complications with advancing age [17]. Our findings corroborate this as we saw higher rates of comorbidities including congestive heart failure, chronic pulmonary disease, and hypertension among the elderly cohort. Our findings suggest a higher risk of mortality for each additional year of age in the adult and elderly operative and non-operative populations. The mean age for the adult deceased population was 6 years older than that of the survived group. Similarly, the mean age for the deceased elderly group was 3 years older than that of the survived group. In our analysis, older subjects had higher Modified Frailty Index than younger patients; importantly, research has demonstrated that frailty as measured by the Frailty Index is a significant predictor of mortality generally [18]. Therefore, the higher Modified Frailty Index we found among our older patients may help to explain why increasing age is a risk factor of mortality (Table 2). Sorensen et al. suggest that the male sex and age are risk factors for postoperative wound complications [19]. Similar to other studies, our results show that men are more common in the surgery group than women, and older age predicts mortality [6,12,15,20]. Khorgami et al. conducted a retrospective analysis from 2008–2014 data of 103,635 patients who underwent elective repair for ventral hernia, 63,685 of whom had the diagnosis of umbilical hernia. They found that older age, male gender, congestive heart failure, pulmonary circulation disorder, coagulopathy, liver disease, fluid and electrolyte disturbances, metastatic cancer, neurological disorders, and paralysis were associated with mortality during hospitalization [15]. Our results concur with their findings, with a few exceptions, as we did not find a significant difference in either the number of adults with congestive heart failure or paralysis or elderly patients with pulmonary circulation disorder, in the survived versus deceased group. Therefore, caregivers should remain vigilant for complications of comorbidities when treating older patients with UH.

### 4.2. Payment Methods and Mortality

Our findings suggest Medicare, Medicaid and self-pay payment models are a predictor for mortality in adults that underwent operation, and private insurance is a protective factor. The adult population had a mortality rate of 0.3% for private insurance, and a mortality rate of 1.2% for Medicare, Medicaid, or self-payment (Table 3). This trend did not hold true in the elderly population, likely because most patients were enrolled in Medicare.

In a retrospective analysis of NIS 2009–2013, Mehta et al. concluded that self-pay patients were more likely to be emergently admitted for inguinal hernia repair which had a higher associated rate of mortality [21]. LaPar et al. note similar findings such that Medicaid and uninsured payer status increases in-hospital mortality in major surgical operations by 97% and 74%, respectively, while private insurance patients had the lowest mortality independent of the operation [22]. Bowman et al. conducted a retrospective medical record review of 321 patients, and identified that compared to privately insured patients, patients with Medicaid were more likely to present with incarcerated or strangulated ventral hernias, and had longer hospital stays [23]. Specifically, in the adult population, Medicare/Medicaid/self-pay patients were disproportionality seen in the deceased group verses that of the survived group (Table 3). Nunez et al. conducted a large retrospective analysis from the 2006–2014 Nationwide Emergency Department Sample and identified that uninsured, publicly insured, and low income patients were more likely to present to the ED with uncomplicated hernia [24]. The authors of this study suggest there may be a disparity in primary surgical care for non-urgent surgical conditions. These findings may help explain why payment method was only a risk factor in the surgery group, as adult patients who present with uncomplicated hernias may forgo further treatment after diagnosis, allowing the condition to progress and ultimately requiring emergent surgery as the hernia develops complications.

Additionally, diabetes and chronic pulmonary disease were found to be more common in Medicare and Medicaid patient populations which can help explain the higher odds of mortality [22]. This information is important to note as the percentage of Medicaid and self-pay admissions has increased from 2001–2010 and 2010–2014, while that of private insurance has declined [12,24]. The socioeconomic factors that predict mortality are vital to patient management as an easily treatable condition may develop into a life threatening one without proper physician follow-up. The progression from uncomplicated to complicated hernia is out of the scope of this study, but future work is needed to further investigate the possibility that certain populations may be at higher risk due to socioeconomic factors that ultimately help predict mortality. Additionally, these findings coupled with the research regarding the increased risks facing financially disadvantaged patients including ED visits and chronic conditions suggests a larger public health issue is at hand here, and we recommend investigating how to try to reduce these disparities on a large scale.

### 4.3. Hernia Complications and Mortality

Hernia complications were predictors of mortality for the adult and elderly operative groups and the elderly non-operative group. Specifically, the presence of gangrene was a risk factor for mortality in adult and elderly patients who had an operation. Obstruction, on the other hand, was a risk factor for elderly patients who did not have surgery. A very large percentage of patients included in the sample experienced an obstruction, including about 81% of adults and 84% of elderly patients, compared to a relatively modest 2.1% of both adult and elderly patients who experienced gangrene. Our data indicate that elderly non-operative patients who had an obstruction had 4.53-times higher odds of mortality than those who did not. The odds ratio for mortality with gangrene was 2.5-fold greater in the elderly operative sample (13.914) than that of the adult operative sample (5.635). Based on these data, elderly patients have greater odds of mortality due to hernia complications (Table 5 and Table 6). This may be explained in part by the increased age, frailty, or HLOS for elderly patients.

Similarly, a retrospective study of 33,700 elderly patients emergently admitted with ventral hernia found that the presence of gangrene and obstruction were both predictors of mortality [25]. Additionally, the literature shows that an incarcerated or strangulated hernia is a risk factor for obstruction or gangrene, respectively, while it is also a typical indication for emergent hernia repair [4,26,27,28]. Prior studies found that emergent repair of hernia is associated with increased morbidity, mortality, and HLOS [3,12,16,29,30,31]. This highlights the need for early intervention and surveillance to prevent such complications requiring emergency surgery [26,32].

### 4.4. HLOS and Mortality

Increasing HLOS was a predictor for mortality in the adult non-operative sample. Specifically, there was a 7.7% increase in the odds of mortality for each additional day spent in the hospital. Similarly, associations between mortality and HLOS were noted in multiple recent retrospective studies on patients emergently admitted with gastroparesis, hemorrhoids, duodenal ulcers, blunt chest wall trauma, tracheostomy, rectal malignancy, total hip arthroplasty, and paralytic ileus [33,34,35,36,37,38,39,40]. The HLOS was not a significant predictor for mortality in the elderly sample, despite elderly patients experiencing a longer HLOS (Table 2). Elderly deceased patients had a HLOS that was nearly 2-times longer than their surviving counterparts; similarly, adult deceased patients had a HLOS that was more than 3 times that of the survived adult patients (Table 3). HLOS was longer in all patients who underwent surgery; however, it was not a risk factor of mortality within these patients.

Moreover, the Modified Frailty Index is significantly higher in the adult non-operative group than that of the operative group (Table 4); it was also higher in the deceased pool when compared to the survived pool in both age populations. This is related to the fact that adults in the non-operation group had more comorbidities than subjects in the adult surgery group (Table 4) [41]. This may shed light on why a longer HLOS was a risk factor of mortality in non-operative patients. In fact, our data note that, when compared to the surgical group, patients in the non-surgical group had a significantly greater proportion of cases with congestive heart failure, chronic pulmonary disease, coagulopathy, liver disease, fluid/electrolyte disorders, metastatic cancer, neurological disorders, and paralysis (Table 4). These are all risk factors that Khorgami et al. associated with mortality during hospitalization [15].

The literature is scarce regarding HLOS and mortality in UH, but some inferences can be made. Patients in the adult non-operative group likely experienced a greater risk of mortality with a longer HLOS due to the “watchful waiting” approach. The efficacy of this approach is still under question by the literature and warrants further investigation. Watchful waiting may be considered as an alternative for small hernias in high-risk patients such as pregnant women, cirrhotic patients, and elderly patients [26,42]. Verhelst et al. suggest watchful waiting may be used for patients with many comorbidities, obesity, or asymptomatic patients, but conclude that watchful waiting leads to increased mortality compared to the operative group [43]. Studies performed for a similar surgery, such as groin hernia repair, draw concurring conclusions [8,44]. This helps explain our findings that adult non-operative patients may be considered higher risk, based on the presence of more comorbidities, and their vulnerability to additional days in the hospital. Interestingly, our study did not find time to operation to be a risk factor for mortality, contrary to other studies where an increased time to operation increased the odd of mortality [13,25,34,35,38,40]. While there is ample literature studying the different surgical methods for umbilical hernia repair, there is clearly a need to further investigate the impact of delaying surgery, HLOS, and the non-operative “watchful waiting” approach on the mortality of patients with UH. Even fewer have identified risk factors of mortality in this non-operative group and this indicates an area that needs further research.

### 4.5. Frailty and Mortality

The data collected from our study indicate that frailty is associated with mortality in the elderly operative group. Similarly, other studies conclude that frailty is an accurate predictor of morbidity and mortality in patients undergoing surgery, and can thus be a valuable tool in the peri-operative process [45,46]. Increased frailty has been shown to be associated with major complications and readmission [47]. Specifically, our findings concur with three retrospective analyses assessing risk factors for mortality in patients admitted with duodenal ulcers, ventral hernia, and hemorrhoids, which found that frailty increased odds of mortality [25,34,35]. This can be explained by the limited reserve capacity of elderly patients to compensate for stress, metabolic derangement, and drug metabolism [48], which can subsequently increase the risk of adverse events following exposure to stressors including surgery and anesthesia [49]. Our data indicate that the average Modified Frailty Index was 0.96 in the adult population and 1.72 in the elderly population. A greater proportion of adult patients underwent surgery than that of elderly patients. This may be explained by the increased risk of postoperative complications in frail patients, causing surgeons to be reluctant to perform an elective repair, as the incidence of additional pathologic conditions increase with age [26]. The reluctance to perform a surgical hernia repair on elderly patients should be quelled with studies showing that elderly age status should not be a bar to perform elective hernia repair, though every effort should be taken to prevent complications as elderly patients are not well suited to tolerate them [7,49]. Our data show that there was no significant difference in the number of deceased subjects in the elderly surgical versus non-surgical groups—this leads us to believe that operation status is not likely be the reason these patients died, but instead this was due to age and frailty.

### 4.6. Study Strengths

The main strength of this study is the combination approach using both the generalized additive model (GAM) and logistic regression model. The NIS database analyzes about 7 million patient records yearly, which allows for a substantial sample size while conducting this thorough retrospective cohort study. This database allows us to investigate diseases in a novel manner, highlighting disease conditions, optimal care, and patient outcomes. This analysis was done across many domains including insurance status, income quartile, race, sex, age, hospital location, comorbidities, and more. The large administrative database holds great potential in understanding the patterns of care and outcomes for research. Many prior studies assessing factors that increase mortality for umbilical hernia were limited to small population sizes from single hospitals or regional location. This study considers the epidemiology and demographics of umbilical hernia from a large sample size and analyzes their impact on treatment and mortality, which have yet to be examined in conjunction. Therefore, these results are likely generalizable across many different settings.

### 4.7. Study Limitations

Since this is a retrospective study using an administrative database, there were potential confounding variables that were not analyzed in this research. Specifically, future studies would benefit from including the following information in the analysis: cause of death, severity of comorbidities, type of hernia repair, size of hernia, emergent or elective repair, and hernia recurrence. It is necessary for further research to investigate the complexity of cases along with other modifiable patient factors, such as choice of operative approach, that influence patient outcomes. In conclusion, there are still gaps in the understanding of umbilical hernia mortality, and our paper helps broaden this knowledge to further influence and impact the care of patients with UH.

## 5. Conclusions

Patients in the elderly sample were found to have a mortality that was 3.5-times greater than the adult sample. All patients were found to have an increased risk for mortality with increasing age.

## Figures and Tables

**Table 1 ijerph-19-10402-t001:** Procedures of emergently admitted patients with the primary diagnosis of umbilical hernia.

**Surgical Procedure (ICD 9)**
Operations on Esophagus (42.01–42.19, 42.31–42.99)
Operations on Stomach (43.0–44.03, 44.21–44.99)
Operations on Intestine (45.00–45.03, 45.30–46.99)
Operations on Appendix (47.01–47.99)
Operations on Rectum, Rectosigmoid, and Perirectal Tissue (48.0–48.1, 48.31–48.99)
Operations on Anus (49.01–49.12, 49.31–49.99)
Operations on Liver (50.0, 50.21–50.99)
Operations on Gallbladder and Biliary Tract (51.01–51.04, 51.21–51.99)
Operations on Pancreas (52.01–52.09, 52.21–52.99)
Operations on Hernia (53.00–53.9)
Operations on Other Operations on Abdominal Region (54.0–54.19, 54.3–54.99)
**Invasive Diagnostic Procedure (ICD 9)**
Invasive Diagnostic Procedure on Esophagus (42.21–42.29)
Invasive Diagnostic Procedure on Stomach (44.11–44.19)
Invasive Diagnostic Procedure on Intestine (45.11–45.29)
Invasive Diagnostic Procedure on Rectum, Rectosigmoid, and Perirectal Tissue (48.21–48.29)
Invasive Diagnostic Procedure on Anus (49.21–49.29)
Invasive Diagnostic Procedure on Liver (50.11–50.19)
Invasive Diagnostic Procedure on Gallbladder and Biliary Tract (51.10–51.19)
Invasive Diagnostic Procedure on Pancreas (52.11–52.19)
Invasive Diagnostic Procedure on Other Operations on Abdominal Region (54.21–54.29)

**Table 2 ijerph-19-10402-t002:** Emergently admitted patients with the primary diagnosis of umbilical hernia, presented and stratified based on sex categories (NIS 2005–2014).

		Adult, N (%)			Elderly, N (%)		
		Male	Female	*p*	Male	Female	*p*
All Cases		8616 (58.4%)	6136 (41.6%)		2843 (43.8%)	3647 (56.2%)	
Race	White	5070 (66.5%)	2829 (52.5%)	<0.001	1969 (78.5%)	2276 (72.1%)	<0.001
	Black	913 (12.0%)	1207 (22.4%)		220 (8.8%)	354 (11.2%)	
	Hispanic	1260 (16.5%)	1069 (19.8%)		233 (9.3%)	391 (12.4%)	
	Asian/Pacific Islander	63 (0.8%)	49 (0.9%)		28 (1.1%)	33 (1.0%)	
	Native American	80 (1.0%)	51 (0.9%)		12 (0.5%)	17 (0.5%)	
	Other	234 (3.1%)	185 (3.4%)		45 (1.8%)	86 (2.7%)	
Income Quartile	Quartile 1	2537 (30.4%)	2053 (34.4%)	<0.001	757 (27.4%)	990 (27.7%)	0.340
	Quartile 2	2147 (25.7%)	1488 (25.0%)		725 (26.2%)	937 (26.2%)	
	Quartile 3	1959 (23.5%)	1353 (22.7%)		665 (24.1%)	911 (25.5%)	
	Quartile 4	1706 (20.4%)	1066 (17.9%)		616 (22.3%)	737 (20.6%)	
I N S U R A N C E	Private Insurance	3827 (44.5%)	2585 (42.2%)	<0.001	304 (10.7%)	277 (7.6%)	<0.001
	Medicare	1148 (13.4%)	753 (12.3%)		2429 (85.5%)	3207 (88.1%)	
	Medicaid	1466 (17.1%)	1582 (25.8%)		53 (1.9%)	93 (2.6%)	
	Self-Pay	1317 (15.3%)	789 (12.9%)		14 (0.5%)	40 (1.1%)	
	No Charge	145 (1.7%)	106 (1.7%)		3 (0.1%)	2 (0.1%)	
	Other	690 (8.0%)	305 (5.0%)		39 (1.4%)	20 (0.5%)	
Hospital Location	Rural	866 (10.1%)	609 (9.9%)	0.770	360 (12.7%)	465 (12.8%)	0.940
	Urban: Non-Teaching	3464 (40.2%)	2438 (39.7%)		1235 (43.4%)	1597 (43.8%)	
	Urban: Teaching	4286 (49.7%)	3089 (50.3%)		1248 (43.9%)	1585 (43.5%)	
C O M O R B I D I T I E S	AIDS	33 (0.4%)	12 (0.2%)	0.042	1 (0.0%)	0 (0%)	0.440
	Alcohol Abuse	1233 (14.3%)	216 (3.5%)	<0.001	205 (7.2%)	38 (1.0%)	<0.001
	Deficiency Anemias	694 (8.1%)	544 (8.9%)	0.080	395 (13.9%)	567 (15.5%)	0.060
	Rheumatoid Arthritis	64 (0.7%)	117 (1.9%)	<0.001	39 (1.4%)	118 (3.2%)	<0.001
	Chronic Blood Loss	26 (0.3%)	42 (0.7%)	0.001	19 (0.7%)	20 (0.5%)	0.540
	Congestive Heart Failure	421 (4.9%)	218 (3.6%)	<0.001	527 (18.5%)	660 (18.1%)	0.650
	Chronic Pulmonary Disease	1124 (13.0%)	1086 (17.7%)	<0.001	716 (25.2%)	787 (21.6%)	0.001
	Coagulopathy	704 (8.2%)	198 (3.2%)	<0.001	185 (6.5%)	156 (4.3%)	<0.001
	Depression	422 (4.9%)	604 (9.8%)	<0.001	132 (4.6%)	349 (9.6%)	<0.001
	Diabetes, Uncomplicated	1444 (16.8%)	1292 (21.1%)	<0.001	808 (28.4%)	1132 (31.0%)	0.022
	Diabetes, Chronic Complications	172 (2.0%)	138 (2.2%)	0.290	110 (3.9%)	148 (4.1%)	0.700
	Drug Abuse	411 (4.8%)	142 (2.3%)	<0.001	16 (0.6%)	6 (0.2%)	0.006
	Hypertension	3682 (42.7%)	2533 (41.3%)	0.080	1937 (68.1%)	2666 (73.1%)	<0.001
	Hypothyroidism	248 (2.9%)	642 (10.5%)	<0.001	188 (6.6%)	708 (19.4%)	<0.001
	Liver Disease	2037 (23.6%)	592 (9.6%)	<0.001	414 (14.6%)	254 (7.0%)	<0.001
	Lymphoma	32 (0.4%)	9 (0.1%)	0.011	26 (0.9%)	21 (0.6%)	0.110
	Fluid/Electrolyte Disorder	1359 (15.8%)	933 (15.2%)	0.350	704 (24.8%)	1059 (29.0%)	<0.001
	Metastatic Cancer	31 (0.4%)	52 (0.8%)	<0.001	39 (1.4%)	62 (1.7%)	0.290
	Other Neurological Disorders	218 (2.5%)	184 (3.0%)	0.090	153 (5.4%)	255 (7.0%)	0.008
	Obesity	1870 (21.7%)	2151 (35.1%)	<0.001	423 (14.9%)	855 (23.4%)	<0.001
	Paralysis	43 (0.5%)	19 (0.3%)	0.080	43 (1.5%)	43 (1.2%)	0.240
	Peripheral Vascular Disorders	312 (3.6%)	127 (2.1%)	<0.001	304 (10.7%)	274 (7.5%)	<0.001
	Psychoses	221 (2.6%)	208 (3.4%)	0.003	54 (1.9%)	83 (2.3%)	0.300
	Pulmonary Circulation Disorders	99 (1.1%)	60 (1.0%)	0.320	102 (3.6%)	134 (3.7%)	0.850
	Renal Failure	595 (6.9%)	282 (4.6%)	<0.001	495 (17.4%)	435 (11.9%)	<0.001
	Solid Tumor	94 (1.1%)	43 (0.7%)	0.015	68 (2.4%)	59 (1.6%)	0.026
	Peptic Ulcer	1 (0.0%)	3 (0.0%)	0.310	1 (0.0%)	1 (0.0%)	0.999
	Valvular Disease	129 (1.5%)	109 (1.8%)	0.190	213 (7.5%)	301 (8.3%)	0.260
	Weight Loss	234 (2.7%)	102 (1.7%)	<0.001	144 (5.1%)	165 (4.5%)	0.310
Complication	No Gangrene or Obstruction	1323 (15.4%)	1194 (19.5%)	<0.001	394 (13.9%)	443 (12.1%)	0.043
	Gangrene	230 (2.7%)	84 (1.4%)		70 (2.5%)	72 (2.0%)	
	Obstruction	7063 (82.0%)	4858 (79.2%)		2379 (83.7%)	3132 (85.9%)	
Invasive Diagnostic Procedure		352 (4.1%)	230 (3.7%)	0.300	148 (5.2%)	198 (5.4%)	0.690
Surgical Procedure		7523 (87.3%)	5206 (84.8%)	<0.001	2317 (81.5%)	2800 (76.8%)	<0.001
Invasive or Surgical Procedure		7568 (87.8%)	5238 (85.4%)	<0.001	2337 (82.2%)	2841 (77.9%)	<0.001
Deceased		84 (1.0%)	29 (0.5%)	0.001	83 (2.9%)	96 (2.6%)	0.480
		Mean (SD)	Mean (SD)	*p*	Mean (SD)	Mean (SD)	*p*
Age, Years		48.95 (9.61)	46.59 (11.35)	<0.001	73.62 (6.83)	77.31 (7.98)	<0.001
Modified Frailty Index		0.96 (1.02)	0.96 (1.02)	0.850	1.73 (1.13)	1.70 (1.07)	0.290
Time to Invasive Diagnostic Procedure, Days		2.19 (4.05)	1.82 (3.32)	0.290	4.59 (10.56)	3.42 (4.61)	0.210
Time to Surgical Procedure, Days		0.62 (1.86)	0.69 (1.35)	0.017	0.88 (1.57)	1.00 (1.75)	0.015
HLOS, Days		3.61 (6.01)	3.50 (4.41)	0.200	5.26 (6.62)	5.45 (5.46)	0.210
Total Charges, Dollars		33,839 (49,737)	32,167 (47,200)	0.040	44,122 (66,480)	40,996 (46,382)	0.035

**Table 3 ijerph-19-10402-t003:** Characteristics of emergently admitted patients with the primary diagnosis of umbilical hernia. Data were classified according to outcome categories, NIS 2005–2014.

		Adult, N (%)			Elderly, N (%)		
		Survived	Deceased	*p*	Survived	Deceased	*p*
All Cases		14,682 (99.2%)	113 (0.8%)		6312 (97.2%)	179 (2.8%)	
Sex, Female		6106 (41.8%)	29 (25.7%)	0.001	3551 (56.3%)	96 (53.6%)	0.480
Race	White	7825 (60.7%)	66 (67.3%)	0.360	4135 (75.1%)	108 (70.1%)	0.080
	Black	2106 (16.3%)	13 (13.3%)		548 (9.9%)	26 (16.9%)	
	Hispanic	2313 (17.9%)	13 (13.3%)		610 (11.1%)	14 (9.1%)	
	Asian/Pacific Islander	110 (0.9%)	2 (2.0%)		60 (1.1%)	1 (0.6%)	
	Native American	129 (1.0%)	2 (2.0%)		29 (0.5%)	0 (0%)	
	Other	416 (3.2%)	2 (2.0%)		126 (2.3%)	5 (3.2%)	
Income Quartile	Quartile 1	4571 (32.1%)	31 (28.4%)	0.160	1709 (27.7%)	39 (22.7%)	0.460
	Quartile 2	3603 (25.3%)	33 (30.3%)		1611 (26.1%)	51 (29.7%)	
	Quartile 3	3299 (23.2%)	31 (28.4%)		1530 (24.8%)	46 (26.7%)	
	Quartile 4	2770 (19.4%)	14 (12.8%)		1317 (21.4%)	36 (20.9%)	
I N S U R A N C E	Private Insurance	6419 (43.8%)	20 (17.7%)	<0.001	569 (9.0%)	11 (6.1%)	0.740
	Medicare	1868 (12.8%)	31 (27.4%)		5475 (86.9%)	162 (90.5%)	
	Medicaid	3020 (20.6%)	32 (28.3%)		144 (2.3%)	3 (1.7%)	
	Self-Pay	2088 (14.3%)	22 (19.5%)		52 (0.8%)	2 (1.1%)	
	No Charge	251 (1.7%)	0 (0%)		5 (0.1%)	0 (0%)	
	Other	997 (6.8%)	8 (7.1%)		58 (0.9%)	1 (0.6%)	
Hospital Location	Rural	1462 (10.0%)	13 (11.5%)	0.140	805 (12.8%)	20 (11.2%)	0.820
	Urban: Non-Teaching	5898 (40.2%)	35 (31.0%)		2752 (43.6%)	80 (44.7%)	
	Urban: Teaching	7322 (49.9%)	65 (57.5%)		2755 (43.6%)	79 (44.1%)	
C O M O R B I D I T I E S	AIDS	42 (0.3%)	3 (2.7%)	0.005	1 (0.0%)	0 (0%)	0.999
	Alcohol Abuse	1408 (9.6%)	41 (36.3%)	<0.001	237 (3.8%)	6 (3.4%)	0.999
	Deficiency Anemias	1209 (8.2%)	27 (23.9%)	<0.001	930 (14.7%)	32 (17.9%)	0.240
	Rheumatoid Arthritis	180 (1.2%)	1 (0.9%)	0.999	150 (2.4%)	7 (3.9%)	0.190
	Chronic Blood Loss	67 (0.5%)	1 (0.9%)	0.410	39 (0.6%)	0 (0%)	0.630
	Congestive Heart Failure	632 (4.3%)	7 (6.2%)	0.330	1133 (17.9%)	53 (29.6%)	<0.001
	Chronic Pulmonary Disease	2193 (14.9%)	17 (15.0%)	0.980	1462 (23.2%)	40 (22.3%)	0.800
	Coagulopathy	863 (5.9%)	38 (33.6%)	<0.001	313 (5.0%)	28 (15.6%)	<0.001
	Depression	1021 (7.0%)	4 (3.5%)	0.190	476 (7.5%)	5 (2.8%)	0.017
	Diabetes, Uncomplicated	2716 (18.5%)	17 (15.0%)	0.350	1898 (30.1%)	42 (23.5%)	0.060
	Diabetes, Chronic Complications	309 (2.1%)	1 (0.9%)	0.730	249 (3.9%)	9 (5.0%)	0.470
	Drug Abuse	545 (3.7%)	6 (5.3%)	0.370	22 (0.3%)	0 (0%)	0.999
	Hypertension	6172 (42.0%)	44 (38.9%)	0.510	4495 (71.2%)	107 (59.8%)	0.001
	Hypothyroidism	886 (6.0%)	4 (3.5%)	0.330	873 (13.8%)	23 (12.8%)	0.710
	Liver Disease	2547 (17.3%)	77 (68.1%)	<0.001	636 (10.1%)	32 (17.9%)	0.001
	Lymphoma	41 (0.3%)	0 (0%)	0.999	46 (0.7%)	1 (0.6%)	0.999
	Fluid/Electrolyte Disorders	2222 (15.1%)	67 (59.3%)	<0.001	1666 (26.4%)	96 (53.6%)	<0.001
	Metastatic Cancer	78 (0.5%)	5 (4.4%)	<0.001	93 (1.5%)	8 (4.5%)	0.001
	Other Neurological Disorders	393 (2.7%)	9 (8.0%)	0.001	387 (6.1%)	21 (11.7%)	0.002
	Obesity	4010 (27.3%)	17 (15.0%)	0.004	1259 (19.9%)	19 (10.6%)	0.002
	Paralysis	61 (0.4%)	1 (0.9%)	0.380	80 (1.3%)	6 (3.4%)	0.016
	Peripheral Vascular Disorders	425 (2.9%)	14 (12.4%)	<0.001	557 (8.8%)	21 (11.7%)	0.180
	Psychoses	419 (2.9%)	9 (8.0%)	0.001	134 (2.1%)	3 (1.7%)	0.999
	Pulmonary Circulation Disorders	154 (1.0%)	4 (3.5%)	0.033	224 (3.5%)	11 (6.1%)	0.070
	Renal Failure	861 (5.9%)	15 (13.3%)	0.001	886 (14.0%)	44 (24.6%)	<0.001
	Solid Tumor	133 (0.9%)	4 (3.5%)	0.021	113 (1.8%)	14 (7.8%)	<0.001
	Peptic Ulcer	4 (0.0%)	0 (0%)	0.999	2 (0.0%)	0 (0%)	0.999
	Valvular Disease	236 (1.6%)	2 (1.8%)	0.700	490 (7.8%)	22 (12.3%)	0.027
	Weight Loss	311 (2.1%)	21 (18.6%)	<0.001	280 (4.4%)	29 (16.2%)	<0.001
Complication	No Gangrene or Obstruction	2508 (17.1%)	12 (10.6%)	<0.001	829 (13.1%)	8 (4.5%)	<0.001
	Gangrene	303 (2.1%)	11 (9.7%)		123 (1.9%)	19 (10.6%)	
	Obstruction	11,871 (80.9%)	90 (79.6%)		5360 (84.9%)	152 (84.9%)	
Invasive Diagnostic Procedure		573 (3.9%)	8 (7.1%)	0.080	332 (5.3%)	14 (7.8%)	0.130
Surgical Procedure		12,667 (86.3%)	102 (90.3%)	0.220	4981 (78.9%)	137 (76.5%)	0.440
Invasive or Surgical Procedure		12,744 (86.8%)	102 (90.3%)	0.280	5041 (79.9%)	138 (77.1%)	0.360
		Mean (SD)	Mean (SD)	*p*	Mean (SD)	Mean (SD)	*p*
Age, Years		47.90 (10.44)	54.05 (7.25)	<0.001	75.61 (7.68)	78.60 (8.50)	<0.001
Modified Frailty Index		0.96 (1.02)	1.31 (0.94)	<0.001	1.71 (1.10)	1.97 (1.14)	0.002
Time to Invasive Diagnostic Procedure, Days		2.00 (3.74)	5.57 (5.47)	0.140	3.48 (4.43)	14.45 (31.52)	0.280
Time to First Surgical Procedure, Days		0.64 (1.67)	0.99 (1.40)	0.045	0.93 (1.66)	1.20 (2.03)	0.150
HLOS, Days		3.49 (5.25)	11.43 (13.13)	<0.001	5.24 (5.49)	9.83 (14.94)	<0.001
Total Charges, Dollars		32,428 (47,093)	121,889 (116,078)	<0.001	40,791 (52,826)	97,659 (112,181)	<0.001

**Table 4 ijerph-19-10402-t004:** Characteristics of emergently admitted patients with the primary diagnosis of umbilical hernia stratified based on surgical status, NIS 2005–2014. Data were stratified according to surgery status, NIS 2005–2014.

		Adult, N (%)			Elderly, N (%)		
		No Surgery	Surgery	*p*	No Surgery	Surgery	*p*
All Cases		2028 (13.7%)	12,782 (86.3%)		1373 (21.1%)	5120 (78.9%)	
Sex, Female		930 (46.0%)	5206 (40.9%)	<0.001	847 (61.7%)	2800 (54.7%)	<0.001
Race	White	988 (54.9%)	6911 (61.7%)	<0.001	848 (70.4%)	3397 (76.2%)	0.002
	Black	368 (20.4%)	1752 (15.6%)		153 (12.7%)	421 (9.4%)	
	Hispanic	345 (19.2%)	1984 (17.7%)		149 (12.4%)	475 (10.7%)	
	Asian/Pacific Islander	12 (0.7%)	100 (0.9%)		13 (1.1%)	48 (1.1%)	
	Native American	21 (1.2%)	110 (1.0%)		6 (0.5%)	23 (0.5%)	
	Other	67 (3.7%)	352 (3.1%)		35 (2.9%)	96 (2.2%)	
Income Quartile	Quartile 1	698 (35.5%)	3906 (31.5%)	0.001	379 (28.2%)	1369 (27.4%)	0.940
	Quartile 2	483 (24.6%)	3159 (25.5%)		352 (26.2%)	1310 (26.2%)	
	Quartile 3	453 (23.1%)	2881 (23.2%)		331 (24.6%)	1246 (24.9%)	
	Quartile 4	331 (16.8%)	2454 (19.8%)		282 (21.0%)	1072 (21.5%)	
I N S U R A N C E	Private Insurance	658 (32.5%)	5786 (45.4%)	<0.001	91 (6.6%)	490 (9.6%)	0.003
	Medicare	389 (19.2%)	1512 (11.9%)		1212 (88.3%)	4426 (86.6%)	
	Medicaid	557 (27.5%)	2500 (19.6%)		43 (3.1%)	104 (2.0%)	
	Self-Pay	267 (13.2%)	1844 (14.5%)		13 (0.9%)	41 (0.8%)	
	No Charge	32 (1.6%)	219 (1.7%)		0 (0%)	5 (0.1%)	
	Other	119 (5.9%)	888 (7.0%)		13 (0.9%)	46 (0.9%)	
Hospital Location	Rural	207 (10.2%)	1269 (9.9%)	0.810	181 (13.2%)	644 (12.6%)	0.200
	Urban: Non-Teaching	801 (39.5%)	5139 (40.2%)		622 (45.3%)	2212 (43.2%)	
	Urban: Teaching	1020 (50.3%)	6374 (49.9%)		570 (41.5%)	2264 (44.2%)	
C O M O R B I D I T I E S	AIDS	15 (0.7%)	30 (0.2%)	<0.001	0 (0%)	1 (0.0%)	0.999
	Alcohol Abuse	242 (11.9%)	1208 (9.5%)	<0.001	45 (3.3%)	198 (3.9%)	0.310
	Deficiency Anemias	238 (11.7%)	1000 (7.8%)	<0.001	249 (18.1%)	713 (13.9%)	<0.001
	Rheumatoid Arthritis	37 (1.8%)	145 (1.1%)	0.009	40 (2.9%)	117 (2.3%)	0.180
	Chronic Blood Loss	10 (0.5%)	58 (0.5%)	0.810	5 (0.4%)	34 (0.7%)	0.200
	Congestive Heart Failure	146 (7.2%)	493 (3.9%)	<0.001	274 (20.0%)	913 (17.8%)	0.070
	Chronic Pulmonary Disease	378 (18.6%)	1832 (14.3%)	<0.001	287 (20.9%)	1216 (23.8%)	0.026
	Coagulopathy	159 (7.8%)	743 (5.8%)	<0.001	75 (5.5%)	266 (5.2%)	0.690
	Depression	166 (8.2%)	860 (6.7%)	0.016	117 (8.5%)	364 (7.1%)	0.080
	Diabetes, Uncomplicated	482 (23.8%)	2254 (17.6%)	<0.001	433 (31.5%)	1507 (29.4%)	0.130
	Diabetes, Chronic Complications	70 (3.5%)	240 (1.9%)	<0.001	57 (4.2%)	201 (3.9%)	0.700
	Drug Abuse	114 (5.6%)	439 (3.4%)	<0.001	5 (0.4%)	17 (0.3%)	0.860
	Hypertension	942 (46.4%)	5276 (41.3%)	<0.001	986 (71.8%)	3617 (70.6%)	0.400
	Hypothyroidism	145 (7.1%)	745 (5.8%)	0.020	195 (14.2%)	701 (13.7%)	0.630
	Liver Disease	493 (24.3%)	2136 (16.7%)	<0.001	143 (10.4%)	525 (10.3%)	0.860
	Lymphoma	7 (0.3%)	34 (0.3%)	0.530	15 (1.1%)	32 (0.6%)	0.070
	Fluid/Electrolyte Disorders	371 (18.3%)	1921 (15.0%)	<0.001	377 (27.5%)	1386 (27.1%)	0.770
	Metastatic Cancer	24 (1.2%)	59 (0.5%)	<0.001	25 (1.8%)	76 (1.5%)	0.370
	Other Neurological Disorders	77 (3.8%)	325 (2.5%)	0.001	103 (7.5%)	305 (6.0%)	0.036
	Obesity	539 (26.6%)	3489 (27.3%)	0.500	232 (16.9%)	1046 (20.4%)	0.003
	Paralysis	16 (0.8%)	46 (0.4%)	0.005	13 (0.9%)	73 (1.4%)	0.170
	Peripheral Vascular Disorders	51 (2.5%)	388 (3.0%)	0.200	154 (11.2%)	424 (8.3%)	0.001
	Psychoses	86 (4.2%)	343 (2.7%)	<0.001	30 (2.2%)	107 (2.1%)	0.830
	Pulmonary Circulation Disorders	38 (1.9%)	121 (0.9%)	<0.001	47 (3.4%)	189 (3.7%)	0.640
	Renal Failure	170 (8.4%)	707 (5.5%)	<0.001	215 (15.7%)	715 (14.0%)	0.110
	Solid Tumor	38 (1.9%)	99 (0.8%)	<0.001	32 (2.3%)	95 (1.9%)	0.260
	Peptic Ulcer	1 (0.0%)	3 (0.0%)	0.450	0 (0%)	2 (0.0%)	0.999
	Valvular Disease	44 (2.2%)	194 (1.5%)	0.030	118 (8.6%)	396 (7.7%)	0.300
	Weight Loss	41 (2.0%)	295 (.3%)	0.420	43 (3.1%)	266 (5.2%)	0.001
Complication	No Gangrene or Obstruction	807 (39.8%)	1715 (13.4%)	<0.001	349 (25.4%)	488 (9.5%)	<0.001
	Gangrene	4 (0.2%)	311 (2.4%)		4 (0.3%)	138 (2.7%)	
	Obstruction	1217 (60.0%)	10,756 (84.1%)		1020 (74.3%)	4494 (87.8%)	
Invasive Diagnostic Procedure		77 (3.8%)	506 (4.0%)	0.730	61 (4.4%)	285 (5.6%)	0.100
Deceased		11 (0.5%)	102 (0.8%)	0.220	42 (3.1%)	137 (2.7%)	0.440
		Mean (SD)	Mean (SD)	*p*	Mean (SD)	Mean (SD)	*p*
Age, Years		48.91 (10.31)	47.79 (10.45)	<0.001	77.65 (8.35)	75.17 (7.45)	<0.001
Modified Frailty Index		1.19 (1.11)	0.92 (1.00)	<0.001	1.74 (1.09)	1.70 (1.10)	0.230
Time to Invasive Diagnostic Procedure, Days		2.11 (1.26)	2.04 (4.05)	0.750	2.50 (1.56)	4.26 (8.53)	0.004
HLOS, Days		2.33 (2.81)	3.75 (5.67)	<0.001	3.25 (3.05)	5.93 (6.45)	<0.001
Total Charges, Dollars		17,302 (20,408)	35,664 (51,291)	<0.001	20,930 (20,615)	48,164 (61,034)	<0.001

**Table 5 ijerph-19-10402-t005:** Backward logistic regression analysis to evaluate the associations between mortality and different factors in emergently admitted patients with the primary diagnosis of umbilical hernia and undergoing an operation. Mortality was the dependent variable. NIS 2005–2014.

		Adult Patients with Operation		Elderly Patients with Operation	
		N = 12,683	R^2^ = 0.095	N = 5118	R^2^ = 0.053
		OR (95% CI)	*p*	OR (95% CI)	*p*
Age, Years		1.066 (1.040, 1.093)	<0.001	1.034 (1.011, 1.057)	0.003
Hernia Complication	No Complication [Ref]	1	<0.001	1	<0.001
	Gangrene	5.635 (2.288, 13.874)	<0.001	13.914 (5.074, 38.154)	<0.001
	Obstruction	1.416 (0.707, 2.834)	0.330	2.269 (0.920, 5.595)	0.080
Insurance	Private Insurance [Ref]	1	<0.001	Removed Via Backward Elimination	
	Medicare	5.440 (3.000, 9.863)	<0.001		
	Medicaid	5.018 (2.775, 9.075)	<0.001		
	Self-Pay	4.575 (2.395, 8.738)	<0.001		
	No Charge	0 (0)	0.995		
	Other	1.803 (0.665, 4.888)	0.250		
Sex, Female		0.547 (0.351, 0.854)	0.008		
Modified Frailty Index		Removed Via Backward Elimination		1.284 (1.105, 1.491)	0.001
Time to Operation, Days				Removed Via Backward Elimination	
Invasive Diagnostic Procedure					
Race					
Income Quartile					
Hospital Location					

**Table 6 ijerph-19-10402-t006:** Backward logistic regression analysis to evaluate the associations between mortality and different factors in emergently admitted patients with the primary diagnosis of umbilical hernia and not undergoing an operation. Mortality was the dependent variable. NIS 2005–2014.

		Adult Patients without Operation		Elderly Patients without Operation	
		N = 2026	R^2^ = 0.139	N = 1373	R^2^ = 0.087
		OR (95% CI)	*p*	OR (95% CI)	*p*
Age, Years		1.213 (1.066, 1.380)	0.003	1.087 (1.045, 1.131)	<0.001
HLOS, Days		1.077 (1.004, 1.155)	0.038	Removed	
Hernia Complication	No Complication [Ref]	Removed Via Backward Elimination		1	0.044
	Gangrene			0 (0)	0.999
	Obstruction			4.534 (1.387, 14.819)	0.012
Sex, Female				Removed Via Backward Elimination	
Invasive Diagnostic Procedure					
Modified Frailty Index					
Race					
Income Quartile					
Insurance					
Hospital Location					

## Data Availability

Data will be available upon request.

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
