# Peer review of "Chances of Mortality Are 3.5-Times Greater in Elderly Patients with Umbilical Hernia Than in Adult Patients: An Analysis of 21,242 Patients"

_ijerph, 2022, doi:10.3390/ijerph191610402_

Round 1

Reviewer 1 Report

Hi

The Article is well designed & written. It's better to add some information a bout etiology of U.H. in analysis

Author Response

Dear IJERPH Editors and Reviewers,

Thank you for taking the time to review our manuscript and provide suggested revisions. The

following changes have been applied:

  1. Page 2: Lines 54-58 – more background on etiology of UH was provided as suggested by reviewer 1.
  2. Page 2: Lines 63-64 – we further clarified the purpose and aim of this article and the unique contribution it will make to the literature as advised by reviewer 2.
  3. Page 2: Lines 85-86 – the income quartiles were explained as recommended by reviewer 2.
  4. Page 3: Line 110 – minor typographical editing was done.
  5. Page 5: lines 125-127 – excessive information was removed. This information did not significantly contribute to the analysis, it can still be found in the tables. Suggested by reviewer 2.
  6. Page 5: Lines 149-151 – we made the results more succinct, removing any excess wordiness for ease of reading. This was suggested by reviewer 2.
  7. Page 7: Line 188 – we made the results more succinct, removing any excess wordiness for ease of reading. This was suggested by reviewer 2.
  8. Page 7: Lines 196-198 – further edits of removing excess information were made; we removed any stats that did not seem to contribute to the purpose of this paper – these stats can still be seen in the tables.
  9. Page 7: Lines 202 – we made the results more succinct, removing any excess wordiness for ease of reading. This was suggested by reviewer 2.
  10. Page 7: Lines 204 – we made the results more succinct, removing any excess wordiness for ease of reading. This was suggested by reviewer 2.
  11. Pages 10-11: Lines 296-298 – we clarified our findings about mortality and comorbidities, as suggested by reviewer 2.
  12. Page 11: Lines 299-303 – excessive information was removed, and the remaining sentences were made more succinct to improve the ease of reading.
  13. Page 11: Lines 303-306 – we added a sentence and a new source to support our findings about frailty, this was suggested by reviewer 2.
  14. Page 11: Lines 323-324 – further edits were made to clean up the wordiness of the discussion. We made sure to only include information that was vital to the paper and removed the excess information.
  15. Page 11: line 337 – excessive numbers were removed – this will make the paper easier to read, suggested by reviewer 2.
  16. Page 12: lines 379-383 – comments were made about the impact of the socioeconomic disparities of UH and how public health issues can be raised. This was suggested by reviewer 2.
  17. Page 12: 416 – minor typographical editing was done.
  18. Page 13: lines 440-442 – we added 4 sources that have a finding that was different from ours, this creates a more critical analysis as disagreements amongst the literature have been brought up. This was suggested by reviewer 3.
  19. Page 13: Lines 443-446 – we suggest that there is a new frontier of research that has yet to be pursued. This contributes to adding value of our paper to the literature along with improving our analysis of the data at hand.
  20. Page 13: Lines 452-453 – an additional source was added to further support our findings.

Responses to each reviewer are provided on an additional document. Thank you again for your

valued feedback and suggestions for improving our article.

Sincerely,

The Authors.

Reviewer 2 Report

Tittle: 

 Chances of Mortality are 3.5 Times Greater in Elderly Patients with Umbilical Hernia than in Adult Patients: An Analysis of  21,242 Patients?

Version: July 30th , 2022.  

Int. J. Environ. Res. Public Health

Reviewer's report:

This is retrospective cohort study with data from   National Inpatient Sample (NIS) 2005-2014 with individuals aged 45 years with aiming to identify risk factors associated with mortality in adult and elderly patients who were hospitalized for umbilical hernia (UH).  As authors highlighted the literature on predictors of mortality is scarce.   The topic of the manuscript is appropriate for the Journal. It could be of interest to investigators and clinicians.  However, several difficulties in sections were identified. As consequence, a major revision is necessary. 

Major compulsory revisions

The first question about this manuscript is… Why the interest to highlight in the tittle their results about this study?  It is no common to introduce numbers of methods or results in the tittle… as consequence… please provide the reasons for this tittle. 

What add this manuscript to information available about UH in the literature? . Authors should insist on different sections about their aim. 

Why the need to explain extensively each table?. Revise the main text and compare with tables is strong recommended to insist in variables related with mortality. 

Why any  comment about delay of surgery?  It is an important factor associated with mortality in older people as authors insisted in reference 23. Is it a limitation? Please clarify. 

Minor essential revisions

Tittle: 
Please see above. 

Abstract:

The abstract is concise and specific.  Purpose of the study is clearly presented. 

Introduction:

The background of the study is clear and precise the problem to be solved.  

The purpose of the article clearly presented. 

Methods:

Sufficient details about the process are provided.  Statistical analyses used are appropriate.  The methods are appropriate and well described. 

The use of multivariable generalized additive model (GAM) to evaluate the role of different factors in anticipating mortality is a strength of this paper.

Income quartile is difficult to read. What is exactly means each quartile? Please clarify this for readers. 

Results:

Information is clearly provided. 

Discussion:

The authors did a nice job in the discussion providing important  gaps  in research about UH in older people. However, why the interest to include data comes from results section here? Only very important percentage should be included in discussion. Please revise this situation.

Please see the comments above. 

·      Page 11 Line 258  Authors have the opportunity to clarify this with their results about comorbidities and mortality. It is no clear the use of “could be attributed to the increased incidence o cardiovascular and pulmonary complications…” What say their data?

·      Page 11 Line 266  Please provide the reference to support the comment about frailty and mortality

At the same time, some comment about the recommendations for public health policies is strong desirable. 

The conclusions are logically valid and justified by the evidence presented in the references cited. 

References:

There are 41 and all are appropriate and relevant. 

References up to date.

Tables and figures:

Five tables are shown. They are well presented. However, check the excessive information provided by them. 

Thanks for letting me review this manuscript.
This could be a nice paper. 
Level of interest: An article whose findings are important to those with closely related research interests. 
Quality of written English: Well. 
Statistical review: No. 
Declaration of competing interests:
I declare that I have no competing interest.

Author Response

(The authors gave the same response as above.)

Reviewer 3 Report

The authors aimed to identify risk factors that associate with mortality in adult and elderly 19 patients who were hospitalized for umbilical hernia.

Overall considered,  the manuscript is quite well written and structured but Discussion section should be expanded with more critical details. In the current format discussion section is a little superficial.

areas of the strength: study well-conceived, appropriate design with the combination approach using both the generalized additive model (GAM) and logistic regression model. Good insight into risk factors of UH development. 

areas of weakness: discussion is too superficial and weak. More recent papers on the topics of the article need to be found and discussed in a more critically way.

Author Response

(The authors gave the same response as above.)

Round 2

Reviewer 2 Report

All comments have been satisfactorily addressed , with thanks. I have nothing further to add. With kind regards.

Reviewer 3 Report

Amended manuscript is now acceptable